# Direct Measuring Particulate Matters in Smoke Plumes from Chimneys in a Textile Dyeing Industrial Park by a Self-Developed PM Detector on an UAV in Yangtze River Delta of China

**DOI:** 10.3390/s22124330

**Published:** 2022-06-08

**Authors:** Zhentao Wu, Xiaobing Pang, Zhangliang Han, Kaibin Yuan, Shang Dai, Jingjing Li, Jianmeng Chen, Bo Xing

**Affiliations:** 1College of Environment, Zhejiang University of Technology, Hangzhou 310014, China; 2112027144@zjut.edu.cn (Z.W.); hanz@zjut.edu.cn (Z.H.); 2112027132@zjut.edu.cn (K.Y.); 2111927039@zjut.edu.cn (S.D.); jchen@zjut.edu.cn (J.C.); 2Shaoxing Ecological and Environmental Monitoring Center of Zhejiang Province, Shaoxing 312000, China; lijjsx@163.com (J.L.); bobo_xing@163.com (B.X.)

**Keywords:** particulate matter, textile dyeing industry, printing and dyeing process, smoke plume, unmanned aerial vehicle

## Abstract

Directly measuring particulate matters (PM) from chimneys in an industrial park is difficult due to it being hard to reach the peak heights. A self-developed PM detector on an unmanned aerial vehicle (UAV) had been deployed to directly measure the PM emissions in smoke plumes from chimneys in a textile dyeing industrial park. Compared with a commercial PM device (LD-5R, SIBATA, Kyoto, Japan), the self-developed detector showed similar performance with a good correlation (R^2^ varying from 0.911 to 0.951) in simultaneously vertical PM measurements on UAV. The PM emissions from chimneys after different textile treating processes, including pigment printing, dyeing process, and digital printing, were investigated. PM mass concentrations and particle number concentrations (PNC) in different sizes were found to be significantly higher in pigment printing than those in dyeing process and digital printing by 2 or 3 times after electrostatic precipitation. The activated carbon adsorption and electrostatic precipitation were the major PM controlling techniques in the park. The PM mass concentrations and PNC were the highest in the process of dyeing after activated carbon adsorption with the concentrations of PM_1_ (1000 μg·m^−3^), PM_2.5_ (1600 μg·m^−3^), and PM_10_ (2000 μg·m^−3^), respectively. According to the results of PM and PNC, PM_2.5_ was found to be the dominant particles accounting for 99% of the PM emissions. It may be due to the high temperature in thermo-fixing machine, which is beneficial to the PM_2.5_ generation. This study revealed PM_2.5_ was the dominant particles to be reduced in textile dyeing enterprises to mitigate PM pollution.

## 1. Introduction

In textile dyeing industrial park a great number of particulate matters (PM) with toxic substances are released into ambient air, which causes serious environmental pollution [1,2]. In 2020, chemical fiber production in China was approximately 49.2 million tons, which accounted for more than 70% of the world’s total output [3]. Shaoxing, a city in the Yangtze River Delta of China, is a center of the textile dyeing industry in China with about 40% of the domestic annual production capacity [4].

The PM emissions are difficult to directly measure from the chimneys in industrial park due to the peak heights being hard to reach, which makes the manual sampling extremely dangerous and time-consuming. Most studies on PM emissions in industrial parks have been conducted at some static sites on ground [5,6,7,8,9]. PM emissions from a coal-fired power plant in southeast Portland were found to be significant sources of sulfate and mercury emissions based on x-ray fluorescence analysis on the ground [10]. The particle size distribution, metals, and polycyclic aromatic hydrocarbons (PAHs) content in PM from five industrial enterprises (cement, chemical, thermal power plant, sponge-iron, and steel) in Tamil Nadu, India, were measured and the emission sources for different polluted components were successfully traced [11]. The PM_2.5_ concentrations from the indoor air of the manufacturing department in a metal factory ranged between 86.3 and 404.9 μg·m^−3^ in Istanbul, measured by a portable PM detector [12]. The above studies are all based on ground measurement, which investigate the PM emissions, and there are only a few studies on PM emissions at high altitudes. The concentrations of PM over the fugitive sources in Moravian-Silesian metropolitan area were significantly higher in spring than in summer by a helium-filled balloon with measuring instrumentation [13]. However, the helium-filled balloons present some difficulties, including the need for a large ground-based crew, safety considerations, and limited freedom of movement [14]. The above-mentioned techniques are unsuitable for directly detecting PM emissions in smoke plumes from chimneys in industrial parks due to the difficulty of reaching the chimney top using the heavy detectors on the ground. 

In recent years, a new monitoring technique using low-cost micro-sensors loaded on an unmanned aerial vehicle (UAV) has provided a new solution for rapid detection of aerial emissions in atmospheric science [15,16,17,18]. The dust (DSM501A) sensor was used to design a UAV-based PM_2.5_ air quality monitoring system [19]. The original PM_2.5_ distribution was study by a multi-sensor system on a UAV in Japan [20]. Recently, the three-dimensional distribution of PM concentration at an institute of water-saving agriculture was measured by sensor-equipped UAV [21]. The PM_2.5_ concentrations along city main roads in Nanjing, China, were quantified by a UAV [22]. The vertical profile of particle number concentrations (PNC) adjacent to a motorway was determined using an UAV [23]. However, little research had been conducted to directly measure the PM emissions from industrial park, especially from the chimneys with smoke plume. Moreover, the pollution characteristics of exhaust between chimneys and fugitive emission in the same workshop were significant different [24]. Identification of PM characteristic under different production would help to accurately assess the source of pollution in textile dyeing enterprises.

In this study, a self-developed PM detector loaded at an UAV was applied to measure the PM from the chimneys of four textile dyeing enterprises in an industrial park. Its accuracy was compared with a commercial PM detector at different heights and smoke plumes. The aim of the study was not only to develop a novel measurement system, but also to investigate the PM emissions from different textile treating processes.

## 2. Materials and Methods

### 2.1. Sampling Locations

UAV measuring campaigns were conducted in a textile dyeing industrial park in Shaoxing city in Yangtze River Delta of China on 30 and 31 January 2021. The detailed location is shown in Figure 1. Four textile dyeing enterprises (named as A, B, C, and D) were selected as the sampling sites since they were the major factories in the park. Three main processes were used for the textile dyeing enterprises, including dyeing, pigment printing, and digital printing. A large amount of exhaust was emitted through the chimneys from the above processes in finishing workshop. In order to clearly investigate the effect of the above treating processes on PM emissions, the PM mass concentration and PNC in the smoke plume were measured directly at eleven chimney tops (A1–A3, B1–B4, C1–C3, and D1) from different textile treating processes. The detailed process of exhaust from the eleven chimneys in four textile dyeing enterprises is shown in Table 1.

### 2.2. PM Detectors and UAV

A self-developed PM detector was employed to measure PM, consisting of a PM sensor (OPC-N2, alphasense, Braintree, UK) and a sampling pump with a size of 22 cm (length) × 15 cm (width) × 10 cm (height). Its weight was 1.0 kg, including a lithium battery for over 8 h operation. The PM sensor (OPC-N2, alphasense, Braintree, UK) used in this study has a size of 7.5 cm (length) × 6 cm (width) × 6 cm (height), a mass of 105 g, a power consumption of 5 w, a measurement range of 0–10,000, and a detection limit of 10. It has a separate inlet port about 5 cm from the bottom of the UAV platform. The data collection frequency is set to 1 s according to software, and the measurement data can be viewed in real time by logging into the cloud platform. A six-rotor UAV was selected to carry PM detector in this field campaign (Figure 2), with a loading platform installed at the UAV bottom. The detailed descriptions of the detector and the UAV can be seen in our previous work [25].

In order to check the performances of the self-developed PM detector, a commercial particle device (LD-5R, SIBATA, Kyoto, Japan) was chosen to compare. The commercial particle device can measure PM_2.5_ concentration and is powered by dry cell batteries. The device size was 18.4 (length) × 6.8 (width) × 11 cm (height) and its weight was 1.0 kg. The main components of the commercial particle device consisted of sampling pump, laser source, scattering plate, lens, and photo detector. Both of them were loaded on the UAV to conduct simultaneously vertical observations of PM from 0 m to 500 m in height in an industrial park from 10:00 a.m. to 15:00 p.m. on 30 January 2021.

### 2.3. Experimental Process

Before measuring, the LD-5R device was calibrated using its own calibration function (zero and span) in the laboratory and then mounted together with the self-developed PM detector on the bottom of the UAV. The UAV was maneuvered by a pilot into the smoke plumes (about 30 m above the ground) for PM measurements online with a flight period of 10 min (Figure 3). The locations of UAV were adjusted in order to ensure that the PM detectors were always inside the smoke plume during the measurement process. 

## 3. Results and Discussion

### 3.1. Comparison with Self-Developed PM Detector and Commercial Device

An outdoor PM comparison measurement of the sensor OPC-N2 and Grimm (Model 180, Hamburg, Germany) is shown in Figure 4. Figure 5 shows an analysis of the correlation between both instruments. Grimm is a professional PM device certified by the US-EPA. The sensor data were in good agreement with the Grimm data over the whole campaign period, with R^2^ values varying from 0.802 to 0.831 (*n* = 434).

The results of particle size calibration comparison were presented in the OPC-N2 sensor datasheet. Figure 6 shows the results of particle size derivative comparison. The OPC-N2 sensor had a consistent linear compared to TSI 3330 (Minneapolis, MN, USA) and Grimm 1.108 (Hamburg, Germany). Figure 7 shows the calibration results for 0.75 and 3 µm Polystyrene Latex standard particles by OPC-N2. The sensor measurements showed that the particle sizes were mainly concentrated between 0.75 and 3 µm.

The vertical comparison experiment was carried out at a location 200 m from the chimney due to the presence of obstructions around the chimney that are not suitable for vertical PM measurement. The height of each chimney of the four textile dyeing enterprises in industrial park is about 25 m. The flight height of the experiment was controlled to about 30 m in order to ensure that UAV sampled in smoke plumes. Figure 8 shows the PM_2.5_ results with the correlation analysis between self-developed PM detector and commercial device in the industrial park at 10:00 a.m. and 14:00 p.m. on 30 January 2021. It can be seen that there was the same consistency for all the PM monitoring results of self-developed PM detector and the commercial device. Both devices show a good agreement with a linear regression correlation coefficient (R^2^) up to 0.9 at different times, suggesting that the self-developed PM detector is a reliable and accurate instrument for PM measurement. 

The measurement results of self-developed PM detector and commercial device in C and D were compared and are shown in Table 2. It shows that all the measurement values of self-developed PM detector were a little smaller than those of commercial devices, but their Relative Percent Differences (RPD) are between 3% and 9%. The differences may be attributed to the systemic errors since two devices were not calibrated by the standard particles. The detector of laser scatter principle was always influenced by water molecules [26]. The commercial particle device is based on the principle that the intensity of laser scattering is proportional to the mass concentration of PM. The self-developed PM detector determines PM mass concentrations by measuring the amount and size distribution of PM. 

### 3.2. PM Concentrations and Size Distribution in Dyeing or Printing Process

The PM mass concentrations of PM_1_, PM_2.5_, and PM_10_ in the plumes of four textile dyeing enterprises are shown in Figure 9. Compared with pigment and digital printing, the mass concentration of PM_1_, PM_2.5_, and PM_10_ in the dyeing process fluctuated slightly by the time. In contrast, the PM mass concentration of pigment and digital printing all showed an increasing trend over time, especially for PM_2.5_ and PM_10_. It indicates that the PM emission of pigment and digital printing is not stable in smoke plume from the chimneys, which may ease causes of sudden PM_2.5_ pollution.

The PM measurement results (the average PM mass concentration and PNC) of dyeing process (A1, A3, B1, B2, C1, C2, C3, and D1), pigment printing (A2), and digital printing (B3 and B4) are shown in Figure 10.

The PM mass concentration and PNC of chimney D1 were the highest, about 8 times more than other chimneys in Figure 10a with the concentration of PM_1_ (1000 μg·m^−3^ and 215,000 p·cm^−3^), PM_2.5_ (1600 μg·m^−3^ and 230,000 p·cm^−3^), and PM_10_ (2000 μg·m^−3^ and 232,000 p·cm^−3^), respectively. In addition to chimney D1, the PM mass concentration and PNC emitted from other chimneys are similar (Figure 10a). The concentrations of PM_1_, PM_2.5_, and PM_10_ were around 100 μg·m^−3^, 200 μg·m^−3^, and 250 μg·m^−3^, respectively. The PNC of PM_1_, PM_2.5_, and PM_10_ increased from 20,000 to 25,000 p·cm^−3^. As can be seen from Table 1, the activated carbon adsorption is used to remove PM from chimney D1, however, activated carbon can easily reach adsorption saturation [27] which results in a poor PM removal performance compared with other chimneys in Figure 10a. 

The PNCs of PM_1_, PM_2.5_, and PM_10_ increased in turn but to a lesser degree in chimney A2, B3, and B4. The mass concentrations of PM_1_, PM_2.5_, and PM_10_ were about 400 μg·m^−3^, 600 μg·m^−3^, and 800 μg·m^−3^ for chimney A2, respectively. The mass concentrations of chimney B3 and B4 were low compared with chimney A2 with PM_1_ (180 μg·m^−3^), PM_2.5_ (300 μg·m^−3^), and PM_10_ (400 μg·m^−3^), respectively (Figure 10b). This is ascribed to the different combination patterns between raw material and fabric in pigment and digital printing. As shown in Table 1, pigment printing was used in chimney A2 with pigments, and digital printing was used in chimney B3 and B4 with disperse dyes. Pigment printing relies on adhesives and crosslinkers to bind to the fabric, and the affinity is lacking between pigment molecules and fiber molecules [28,29]. The disperse dyes are mainly diffused into the amorphous zone of fabric by van der Waals forces during printing [30]. Thus, the binding action between disperse dyes with fabric is stronger than that between pigment printing with fabric, causing PM to be more easily emitted during the heating process from pigment printing. 

Based on data from Figure 10, the PM_1_, PM_2.5_, and PM_10_ concentrations for digital printing (i.e., B3 and B4) and for pigment printing (i.e., A2) were 1.5–1.8 times and 3–4 times higher than that of the dying process, respectively (i.e., average PM concentrations of chimney A1, A3, B1, B2, C1, C2, and C3). The reactive dyes were used in the dying process, in which molecules contain groups that react with the fabric molecules and bond to the fiber through covalent bonds [31]. The bonding of this material is stronger than that of printing materials, causing the PM from dying products to be harder to release [32]. Pigment printing takes a high temperature heating process of 180–200 °C, and some film-forming substances such as grease, animal, and plant waxes are evaporated by heat [33]. It is easier to form high PM concentration when the hot steam condenses in cold. The temperature of digital printing and dyeing is generally 160–170 °C, which is lower than that of pigment printing. Therefore, the concentration of PM in pigment printing (A2) is higher than that in digital printing (B3 and B4) and dyeing (A1, A3, B1, B2, C1, C2, C3, and D1).

The proportion of four particle size ranges in the plumes of the four textile dyeing enterprises is shown in Figure 11. The dyeing process including A1, A3, B1, B2, C1, C2, C3, and D1 is shown in Figure 11a and the printing process, including A2, B3, and B4, is shown in Figure 11b.

PM distribution sorted from large to small was 0.3–0.5 μm (68.59–72.93%), 0.5–1 μm (20.4–24.39%), 1–2.5 μm (5.36–7.49%), and 2.5–10 μm (0.37–0.72%), respectively. The proportion of PM size distribution below 2.5 μm was found to be over 90%. It may be due to the high temperature, causing oil and wax compounds in the fabric to be volatile in thermo-fixing machine, which is beneficial to the generation of PM_2.5_ [34]. For the dying process (a panel), the sum of proportion of particle size range between 1–2.5 μm and 2.5–10 μm from Chimney D1 is the highest (7.84%), indicating that the end treatment of activated carbon adsorption is harder to capture the large PM than electrostatic precipitation [35], which might cause PM emission higher up (Figure 10a). For the printing process (b panel, A2 pigment printing, B3 and B4 digital printing), the proportion of PM size distribution below 2.5 μm was also the highest. Among the proportion, the PM size of 0.3–0.5 μm was higher than dyeing process (a panel), about 3%.

## 4. Conclusions

A self-developed PM detector with an OPC-N2 PM sensor loaded on an UAV can directly measure PM from smoke plume of chimneys. PM emissions from three main processes in the textile dyeing enterprises, including dyeing, pigment printing, and digital printing, were investigated. The pigment printing was formed by high PM mass and particle number concentrations about 2–3 times higher than digital printing and dyeing. The PM mass concentration or PNC emitted from the processes of digital printing and dyeing were similar. The proportion of PM size range in 0.3 to 0.5 μm was significantly higher in the printing process than in the dyeing process. Overall, the PM_2.5_ was the major emission from the chimneys in all textile dyeing enterprises. Therefore, reducing PM_2.5_ emissions from textile dyeing enterprises, especially from the process of pigment printing, is a priority to control PM pollution in textile dyeing industrial park.

## Figures and Tables

**Figure 1 sensors-22-04330-f001:**
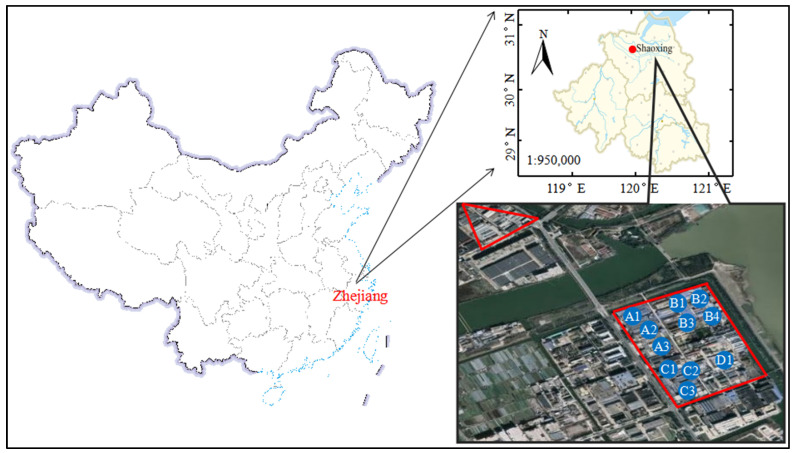
Experimental site: textile dyeing industrial park, Shaoxing, China. The red rectangle square represented the textile dyeing industrial park, and the red triangle represented the resident area. The chimney numbers were pointed in the red rectangle square with blue circles.

**Figure 2 sensors-22-04330-f002:**
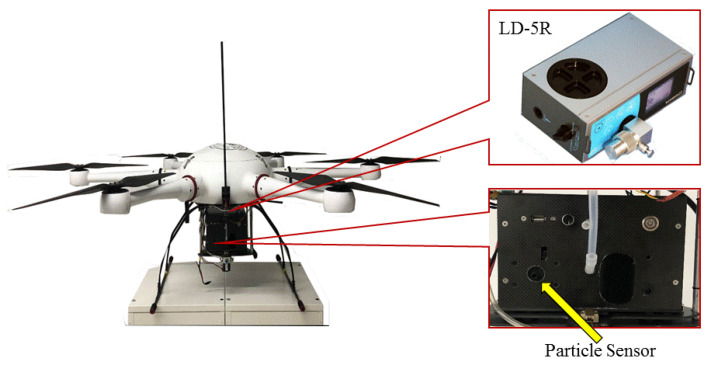
Self-developed PM detector and commercial PM device (LD-5R, SIBATA, Kyoto, Japan) on an UAV deploying in aerial monitoring a textile dyeing industrial park in Shaoxing, Zhejiang province.

**Figure 3 sensors-22-04330-f003:**
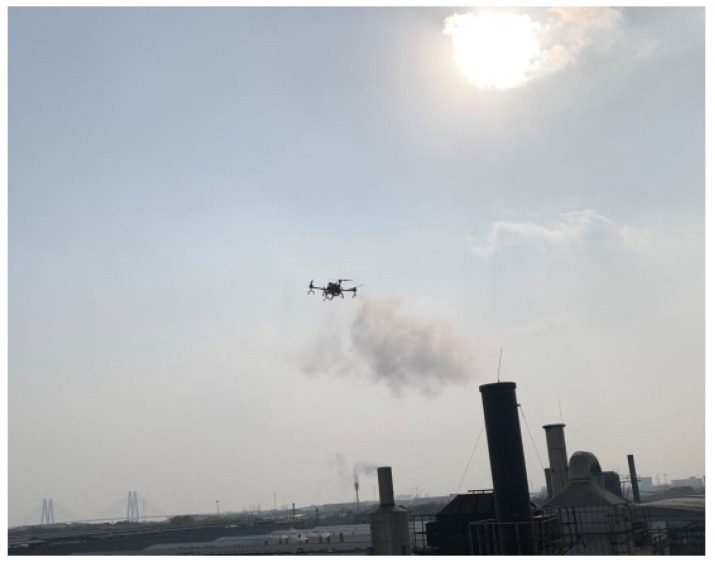
On-situ photograph of PM measurement from the chimney by a self-developed PM detector on an UAV.

**Figure 4 sensors-22-04330-f004:**
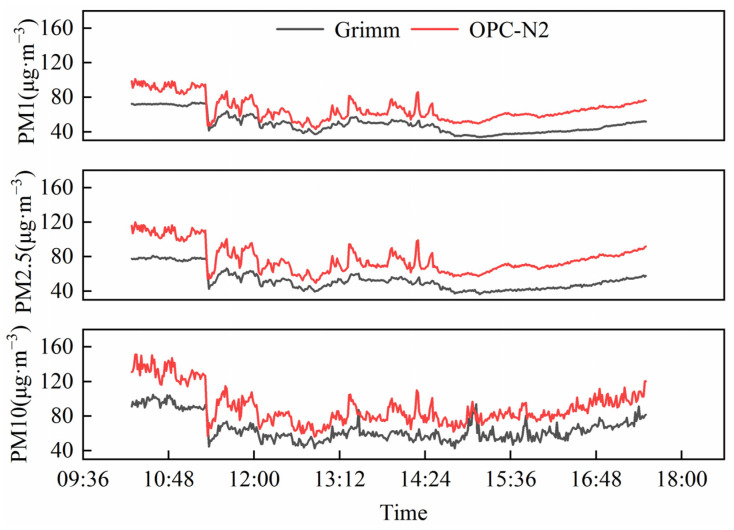
Comparison measurement results of OPC-N2 and Grimm for ambient air PM concentrations in outdoor monitoring.

**Figure 5 sensors-22-04330-f005:**
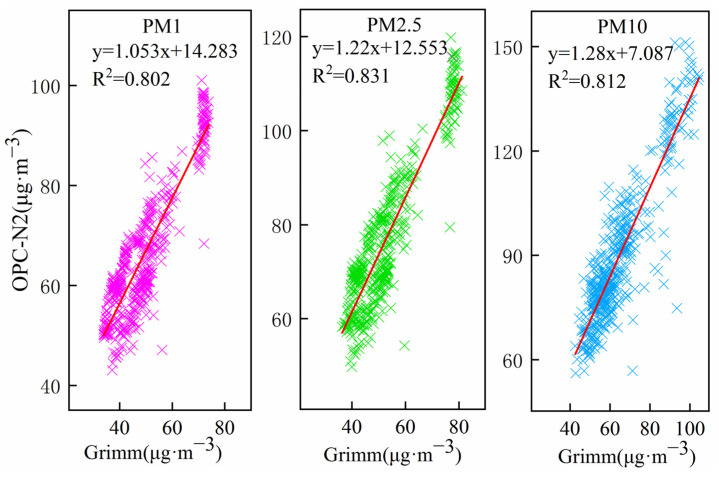
Correlation analysis results of OPC-N2 and Grimm for ambient air PM concentrations in outdoor monitoring.

**Figure 6 sensors-22-04330-f006:**
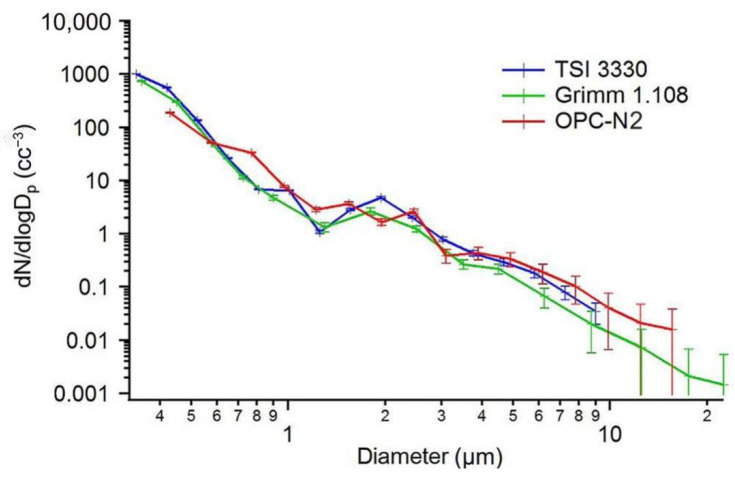
Particle size derivative comparison results of the three PM instruments (OPC-N2, Grimm 1.108, and TSI 3330).

**Figure 7 sensors-22-04330-f007:**
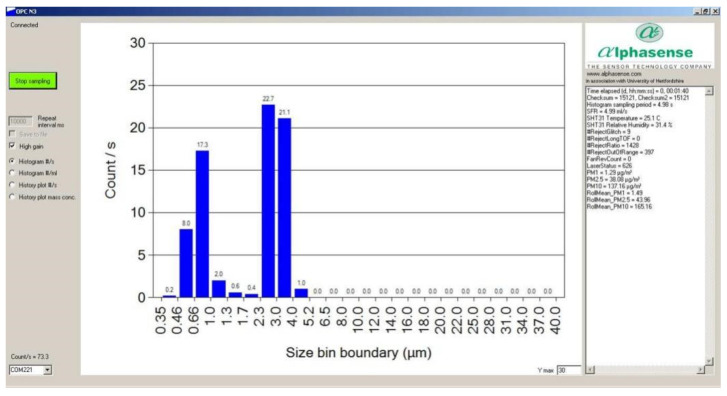
Calibration results for 0.75 and 3 µm Polystyrene Latex standard particles by OPC-N2.

**Figure 8 sensors-22-04330-f008:**
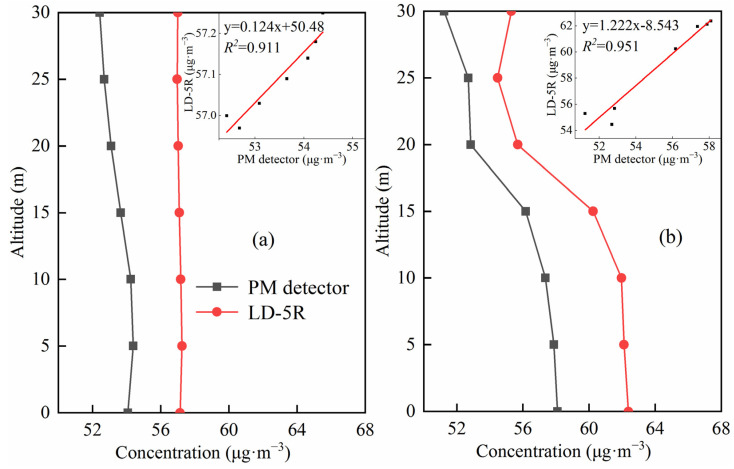
Comparison of PM measurements between self-developed PM detector and LD-5R in vertical observations from 0 to 30 m in 10:00 a.m. ((**a**) panel) and 14:00 p.m. ((**b**) panel) and their correlation analysis in inserted panels.

**Figure 9 sensors-22-04330-f009:**
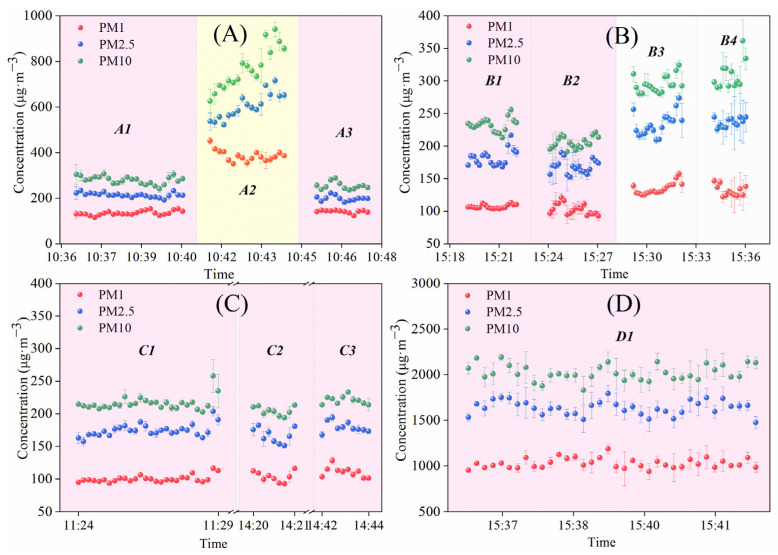
Measurement results of PM concentration in the plumes of four textile dyeing enterprises on 30 ((**C**,**D**) enterprise) and 31 ((**A**,**B**) enterprise) January 2021. A1–A3 represented the number of chimneys in A enterprise. Similarly, B1–B4, C1–C3, and D1 meant the number of chimneys in B, C, and D enterprise, respectively. The background color represented various productions, including dyeing process (pink), pigment printing (yellow), and digital printing (gray) in the four textile dyeing enterprises.

**Figure 10 sensors-22-04330-f010:**
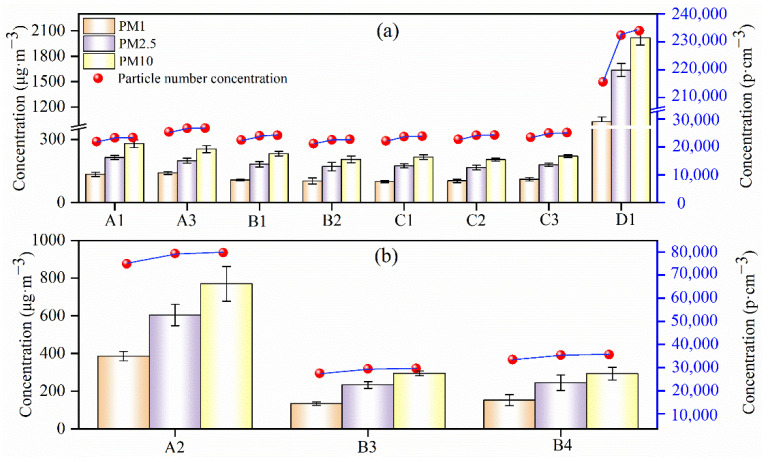
Average PM mass concentration and PNC in the plumes in dyeing process ((**a**) panel) and printing process ((**b**) panel, A2 represent pigment printing, B3 and B4 represent digital printing) of four textile dyeing enterprises (A, B, C, D).

**Figure 11 sensors-22-04330-f011:**
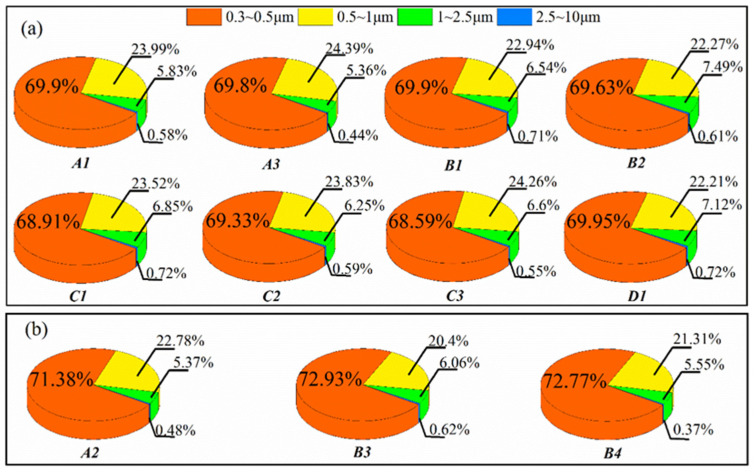
Proportion of four PM size ranges in smoke plumes in dyeing process ((**a**) panel) and printing process ((**b**) panel, A2 pigment printing, B3 and B4 digital printing) of four textile dyeing enterprises (A, B, C, D).

**Table 1 sensors-22-04330-t001:** The detailed process of exhaust from the eleven chimneys in four textile dyeing enterprises.

Chimney No.	Process	Process Description	Exhaust Treatment
**A1, A3, B1, B2, C1, C2 and C3**	Dyeing	coloring fiber with dyes	Electrostatic precipitation
**A2**	Pigment printing	printing fiber with pigments and adhesives	Electrostatic precipitation
**B3 and B4**	Digital printing	printing disperse dyes onto fabric by digital technology	Electrostatic precipitation
**D1**	Dyeing	coloring fiber with dyes	Activated carbon adsorption

**Table 2 sensors-22-04330-t002:** Comparison of average PM_2.5_ concentrations measured between self-developed PM detector and LD-5R commercial device in different sits in industrial park (Unit: μg·m^−3^).

Sites	C1	C2	C3	D1
**LD-5R**	161 ± 10 (*n* = 28 *)*	195 ± 50 (*n* = 10)	197 ± 15 (*n* = 10)	1676 ± 97 (*n* = 36)
**PM detector**	150 ± 15 (*n* = 28)	178 ± 36 (*n* = 10)	189 ± 37 (*n* = 10)	1537 ± 110 (*n* = 36)
**RPD (%)**	6.8	8.7	3.9	8.3

*n* means the number of samples.

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
