# Peer review of "Direct Measuring Particulate Matters in Smoke Plumes from Chimneys in a Textile Dyeing Industrial Park by a Self-Developed PM Detector on an UAV in Yangtze River Delta of China"

_sensors, 2022, doi:10.3390/s22124330_

Round 1

Reviewer 1 Report

The manuscript first developed a new PM Detector, then directly measure the PM emissions from industrial park especially from the chimneys with smoke plume. This work is interesting, however, the contents need to be major revised before acception.

  1. The logic of manuscript needs to be emphasized. If the author wants to emphasis the new PM Detector, then the performance of the new PM Detector should be clearly and entirely presented. If the author wants to emphasis the PM values from industrial park especially from the chimneys with smoke plume, then using LD-5R can be done.
  2. Line 127 The UAV was maneuvered by a pilot into the smoke plumes (about 30 m above the ground) for PM measurements, so the comparison of PM measurements between self-developed PM detector and LD-5R should be focused on 0-30m, not 0-500m on Figure 4. Besides, Figure 4 should point out PM is PM1, PM2.5 or PM10?
  3. Furthermore, the accuracy of PM1 and PM10 should be verified.
  4. On Figure 5 D1, the values of PM1, PM2.5, and PM10 are too large, are they right? Why the values of D1 have lager difference with A1, B1, C1,…etc.
  5. On Figure 7, how are the accuracy of the proportion of four particle size?
  6. The reason of the different PM values in four textile dyeing enterprises should be deeply discussed.

Reviewer 2 Report

1. FromTab.1, the Chimney No.  was not consistent at differeent processes, please explain the reasons.

2. Please explain why  altitude was choosen at 500 meters.

3. The format of Ref.[18] is not correct , please check all the  References .

4. The sensitivity and linearity of sensor should be compared between the  commercial device and the self-developed detector.

5.  The correlation  (R2 = 0.952) was only showen in  abstract. Why?

6. The time range of the experiment is suggested to 24 hours 

Reviewer 3 Report

This paper addresses the use of UAVs to measure air pollution, a popular topic. A brief literature review and description of the research is presented and conclusions are drawn. Some detailed comments follow.

  1. Introduction

There are many articles regarding air quality measurements using UAVs. This section lacked information on measurements using low-cost sensors on UAV platforms. Please complete the information in this area.

  1. Materials and Methods

2.1. Sampling locations

Please label the locations of the stacks in Figure 1, with the symbols in Table 1 Please enlarge the satellite image and reduce the map . This will make it easier to analyze the data.

2.2. PM detectors and UAV

Is the equipment used for the measurement approved for in-flight measurements? The article states the weight of the new device as well as the commercial one to be 1kg, which is a lot for a UAV. Is it possible to reduce the mass of a newly designed device?3.1. Multi-rotor UAV

Why it was decided to use this particular model, was it possible to use smaller UAV platform? What maximum additional load is generated by the installed measurement equipment?

2.3. Experimental process

Has the effect of UAV rotors on the measurements been checked? At what height were measurements conducted? At what distance from the emitter was the measurement conducted? How long was the particle concentration measured at one location?

Were the instrument readings compared to others not in motion?

  1. Results and Discussion

3.1. Comparison with self-developed PM detector and commercial device

Figure 4 shows the vertical distribution of the contamination. The maximum height at which measurements were taken according to the figure is 500m. Were these heights measured at each chimney? How high are the individual chimneys? Please complete the chapter with information on this topic.

3.2. PM concentrations and size distribution in dyeing or printing process

Measurement results are described in quite a detailed way, but in Figure 7 there is no information which graphs refer to which emitters. Please fill it in.

  1. Conclusions

Please provide information about the sensor, at what heights it measures, how it averages the data, is it possible to view the measurement results in real time. Are there any plans to incorporate the sensor into commercial solutions?

Round 2

Reviewer 1 Report

1.     Because the UAV was maneuvered by a pilot into the smoke plumes (about 30 m above the ground) for PM measurements, so the comparison of PM measurements between self-developed PM detector and LD-5R should be focused on 0-30m. From Figure 4, we can find the difference between self-developed PM detector and LD-5R was obvious, so the entire results may be questionable.

2.     The accuracy of PM1 and PM10 was not provided, so the results are not credible.

3.     On Figure 7, because the accuracy of the proportion of four particle size is not provide, so the results are not credible.

Reviewer 2 Report

 English language and style are minor spell check required

Reviewer 3 Report

All reviewer comments have been incorporated into the manuscript. The article is suitable for print in my opinion in its present form.

Author Response

Thanks to the reviewer's suggestion.